# IDENTIFYING TREATMENT RESPONSE SUBGROUPS IN OBSERVATIONAL TIME-TO-EVENT DATA

## ABSTRACT

Identifying patient subgroups with different treatment responses is an important task to inform medical recommendations, guidelines, and the design of future clinical trials. Existing approaches for subgroup analysis primarily rely on Randomised Controlled Trials (RCTs), in which treatment assignment is randomised. RCTs' patient cohorts are often constrained by cost, rendering them not representative of the heterogeneity of patients likely to receive treatment in real-world clinical practice. When applied to observational studies, subgroup analysis approaches suffer from significant statistical biases particularly because of the non-randomisation of treatment. Our work introduces a novel, outcome-guided method for identifying treatment response subgroups in observational studies. Our approach assigns each patient to a subgroup associated with two time-to-event distributions: one under treatment and one under control regime. It hence positions itself in between individualised and average treatment effect estimation. The assumptions of our model result in a simple correction of the statistical bias from treatment non-randomisation through inverse propensity weighting. In experiments, our approach significantly outperforms the current state-of-the-art method for outcome-guided subgroup analysis in both randomised and observational treatment regimes.

## 1 INTRODUCTION

Understanding heterogeneous therapeutic responses between patient subgroups is the core of treatment guidelines and the development of new drugs. Identifying such subgroups is valuable to inform clinical trials by identifying subgroups not responding to existing drugs, and to direct healthcare resources to those who might benefit most, and away from those who may be most harmed (Foster et al., 2011). For instance, subgroup analysis in the BARI trial of patients with coronary artery disease supported the use of coronary artery bypass graft over percutaneous interventions for patients with diabetes, and the opposite for patients without diabetes (Investigators, 1996), shaping subsequent guidelines. Figure 1 illustrates this idea: two groups may present opposite treatment responses, and would benefit from different recommendations. Our work aims to uncover such subgroups.

Randomised controlled trials (RCTs) remain the gold standard for identifying subgroups of treatment effects. In RCTs, patients are randomly assigned to control or treated groups, allowing researchers to assess the impact of an intervention or treatment. However, RCTs are costly and time-consuming, with restricted patient cohorts unrepresentative of the real-world diversity of patients who may receive treatment and treatment strategies (Hernán & Robins, 2016).

Our work introduces a novel approach that diverges from traditional RCT-based methodologies (Nagpal et al., 2022a; 2023) by leveraging routinely collected observational data to identify patient subgroups. Observational studies encompass larger and more diverse cohorts reflective of real-world practices, offering the potential to uncover subgroups of treatment responses, that could be missed in RCTs. Prior works (Bica et al., 2020; Curth et al., 2021; Louizos et al., 2017) have leveraged observational data while addressing biases from non-random treatment assignments (Benson & Hartz, 2000; Hernán & Robins, 2010; Hernán, 2018). However, this body of literature primarily focuses on estimating (i) *averaged* treatment effects at the population level or (ii) *individualised* treatment effects, thereby overlooking the identification of *treatment effect subgroups*.

Addressing this critical gap in the literature, our work uncovers patient subgroups with distinct treatment responses using observational data. Unlike previous methods relying on RCTs, we introduce

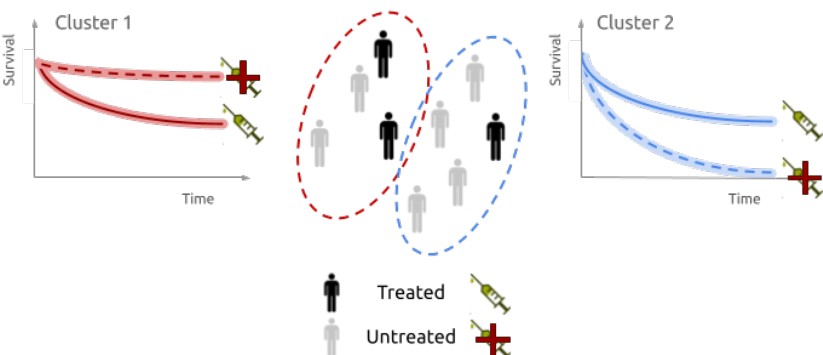

Figure 1: *Subgroup treatment effect discovery in time-to-event observational data.* Our method identifies subgroups of patients with similar treatment responses to guide clinical practice and design clinical trials. Our method simultaneously models the treatment effect and identifies subgroup while addressing censoring and treatment non-randomisation.

a mixture of neural networks as an extension to traditional outcome-guided models to uncover subgroups delineated by non-linear, higher-dimensional combinations of available covariates, without requiring parametrisation of the time-to-event distributions or treatment responses.

**Contributions.** Our work introduces the first neural network approach to simultaneously estimate treatment effects and discover subgroups in *observational* settings, addressing both censoring and non-random treatment assignments without parameterising the survival distribution or treatment effects. Section 2 formalises the problem of treatment subgroup discovery and non-randomisation correction. Then, we introduce the proposed monotonic neural network implementation in Section 2. Finally, we evaluate the proposed methodology on two synthetic datasets in Section 3, an extensive set of alternative settings in App. B and a real-world example described in App. C and compare it with related approaches reviewed in Section 4.

## 2 METHOD

### 2.1 PROBLEM SETUP

**Latent survival subgroup analysis.** Our goal is to uncover subgroups with similar treatment responses, guided by the *observed* times of occurrence of an outcome of interest. Patients are assigned to subgroups based on their covariates at treatment time. Each subgroup is characterised by two distributions, known as survival functions: one under treatment and one under control regimes. These distributions characterise the probability of observing the event of interest after a given time under a given treatment.

Consider the random variables associated with observed covariates $X$, an indicator identifying whether the event of interest, in our analysis death, was observed $D$, and the observed time of event $T$. Formally, we define the latter random variables as $T := \min(C, T')$ and $D := \mathbb{1}(C > T')$, where $C$ is random variable of the (right)-censoring time, i.e. the time at which the patient left the study before experiencing the event of interest, and $T'$ is the partially observed random variable associated with the time of the event of interest if there was no censoring. When a patient is censored, $T = C$ and $D = 0$, otherwise the event is observed and $T = T'$ and $D = 1$. To estimate the associated survival likelihood, we further assume, as commonly done in survival analysis, that the censoring time $C$ is not informative for the time of the event of interest $T'$.

**Assumption 1** (Non-informative censoring)**.** *The censoring time $C$ is independent of the time of the event of interest $T'$ given the covariates $X$. Formally, $T' \perp\!\!\!\perp C \mid X$.*

Central to our problem is the additional variable $Z$ associated with the latent, *unobserved* subgroup membership. Following Jeanselme et al. (2022), we aim to identify a pre-specified $K \in \mathbb{N}$ number of

subgroups[1]. We assume that $Z$ depends on $X$ and influences $T'$. As we are interested in recovering the survival functions associated with the latent subgroups, we ignore the potential dependence of $T'$ on $X$, considering the individual survival distribution as a mixture of the different subgroups.

**Assumption 2** (Mixture modelling). *The event time $T'$ is completely determined by the patient's group membership $Z$. Formally, $T' \perp\!\!\!\perp X \mid Z$.*

While this assumption may hurt individual performance as individual covariates do not directly inform individual survival but group membership only, it improves the interpretability of the model by separating subgroup membership from survival profiles. In conclusion, we aim to recover $Z$ and the survival distributions associated with each subgroup from the observed $X, T$ and $D$.

Given the previously described dependencies, the expected survival $S$ for a given patient with covariates $x$ is the survival when marginalising over the different subgroups:

$$S(t \mid X = x) := \mathbb{P}(T' \geq t \mid X = x) = \sum_{k=1}^{K} \mathbb{P}(Z = k \mid X = x)\mathbb{P}(T' \geq t \mid Z = k) \qquad (1)$$

Note that the last term is linked to the quantity commonly used in the survival literature: $\Lambda_k(t) := -\log \mathbb{P}(T' \geq t \mid Z = k)$, the group-specific cumulative hazard, with derivative with respect to time $\lambda_k(t)$, the group-specific instantaneous hazard, corresponding to the increase in the probability of observing the event of interest given survival until that time $t$. Estimating the two probability distributions $\mathbb{P}(Z = k \mid X = x)$ and $\mathbb{P}(T' \geq t \mid Z = k)$ is our core interest.

**Likelihood-based model fitting.** One can maximise the log-likelihood associated with the observed realisations $\{x_i, t_i, d_i\}$ of the random variables $X, T, D$ for patient $i \in [\![1, N]\!]$ with $N$ the total number of patients (we omit $i$ where redundant) to estimate the survival function (Eq. equation 1). Under Assumption 1, Equation equation 2 describes this *factual*[2] log-likelihood.

$$l_F(\theta) = \sum_{i, d_i = 1} \log \left( -\frac{\partial S_\theta(u \mid x_i)}{\partial u} \Big|_{u = t_i} \right) + \sum_{i, d_i = 0} \log S_\theta(t_i \mid x_i) \qquad (2)$$

where $\theta$ is the set of parameters characterising the estimated survival function $S_\theta$. Here, patients with an observed event ($d_i = 1$) contribute the negative derivative of the survival function to the likelihood (see Jeanselme et al. (2022) for derivation). Each censored patient ($d_i = 0$) contributes the probability of not observing an event before $t_i$, i.e., $S_\theta(t_i \mid x_i)$, under assumption 1 of independence between the censoring time and the event of interest conditional on the covariates.

**Latent treatment effect subgroups.** Consider now the *binary* treatment variable $A$. Patients either receive the treatment ($A = 1$) or they do not ($A = 0$). Therefore, following the potential outcomes formulation, we consider the time of events under treatment $T'_1$ and under the control regime $T'_0$. The central challenge is that one can not observe both $T'_0$ and $T'_1$, but only $T'$, the random variable associated with the event time under the observed treatment regime in the absence of censoring, which is equal to $A \cdot T'_1 + (1 - A) \cdot T'_0$ under assumption 4 described below. Figure 2 describes the dependencies between the previously described random variables.

A critical assumption of our proposed setting is that the subgroups we aim to identify do not influence treatment if we know a patient's covariates, formalised as follows:

**Assumption 3** (Unknown latent groups). *The treatment assignment $A$ is independent of the subgroup membership $Z$ given the covariates $X$. Formally, $A \perp\!\!\!\perp Z \mid X$.*

This assumption relaxes the traditional assumption of treatment randomisation. Note that this assumption remains realistic, as (i) clinicians base treatment recommendations on the patients' covariates, and (ii) if subgroups were known, one would not need the proposed methodology to uncover novel subgroups.

---

[1]As a pre-specified number of subgroups may be a limitation in a real-world setting where we do not know the underlying grouping structure, we explore how to select this parameter based on the likelihood of the predicted outcomes in Section 3.

[2]Referring to the likelihood of the observed (rather than counterfactual) realisations.

Further, the following three assumptions are necessary to estimate treatment effects and are common in the causal literature.

**Assumption 4** (Consistency). *A patient's observed event time is the potential event time associated with the observed treatment. Formally, this means $T' := A \cdot T'_1 + (1 - A) \cdot T'_0$ where $T'$ is the observed event time and $(T'_0, T'_1)$ are the potential event times under the treatment $A$.*

**Assumption 5** (Ignorability). *The potential event times are independent of the treatment given the observed covariates, i.e. $A \perp\!\!\!\perp (T'_0, T'_1) \mid X$. Equivalently, no unobserved confounders impact both treatment and event time.*

**Assumption 6** (Overlap / Positivity). *Each patient has a non-zero probability of receiving the treatment, i.e. $\mathbb{P}(A \mid X) \in (0, 1)$ where $(0, 1)$ is the open interval, resulting in a non-deterministic treatment assignment.*

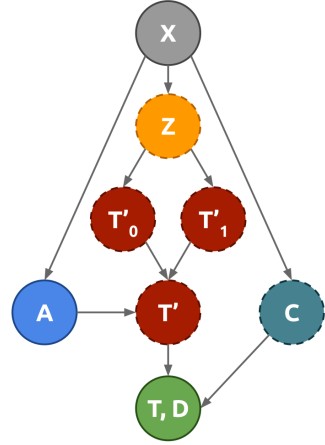

Figure 2: Graphical representation between covariates $(X)$ and outcomes $(T, D)$. Realisations of dashed variables are unobserved, while $X$, $A$, $T$ and $D$ are observed. —

Under the Assumptions 4, 5, and 6, existing works often focus on estimating the Individualised Treatment Effect (ITE, here denoted as $\tau$), defined as the difference between survival under the two potential treatment regimes given the covariates $x$ (see App. A.1 for derivation):

$$\tau(t, x) = \mathbb{P}(T' \geq t \mid A = 1, X = x) - \mathbb{P}(T' \geq t \mid A = 0, X = x) \tag{3}$$

Estimating this quantity requires accurately modelling the survival distributions under the two treatment regimes *for all $x$* from observed data. This estimation would be straightforward if one could access the observed survival times for all patients under both treatment regimes. In this case, one would estimate the survival using the event times under $A = 1$ and under $A = 0$. The key challenge is that the *counterfactual* survival outcome is not observed: if a patient receives the treatment, we do not observe its outcome under no treatment, and vice versa.

In RCTs, treatment $A$ and covariates $X$ are by design independent. Hence, the subset of the patients receiving either treatment regime is representative of the overall population. Relying on this observed subset to estimate the survival distribution under each regime is a valid estimate for the population. In other words, maximising the factual likelihood to estimate survival under both treatment regimes results in a valid estimate. However, covariates and treatment are not independent in observational studies in which treatment recommendations depend upon the observed covariates. Formally, $\mathbb{P}(A \mid X) \neq \mathbb{P}(A)$ in general. For example, clinicians might recommend more aggressive treatment to patients in more severe conditions. This absence of randomisation results in a covariate shift between the treated and non-treated populations (Curth et al., 2021) as their covariate distributions differ (Bica et al., 2021). In this setting, estimating survival by maximising the factual likelihood is no-longer enough to estimate the counterfactual survival distribution.

Under assumption 6, approaches such as re-weighting (Shimodaira, 2000), or penalisation on the dissimilarity of learnt representations (Johansson et al., 2016; Shi et al., 2019), or a combination of these approaches (Curth et al., 2021; Hassanpour & Greiner, 2019; Shalit et al., 2017) are remedies to estimate the counterfactual likelihood from the observed data by reducing the difference between the two treatment regimes' populations when modelling the outcome of interest. Specifically, Shalit et al. (2017) demonstrate that the negative log likelihood is upper-bounded by observable quantities. Here we extend their notations to the survival setting as

$$-l(\theta) \leq -l^*_F(\theta) + \gamma \cdot \text{IPM}(q^{A=0}_\Phi, q^{A=1}_\Phi), \tag{4}$$

where $l$ is the log-likelihood consisting of both factual and counterfactual log-likelihoods, $l^*_F$ is the factual log-likelihood (defined in equation 2) *weighted* with an inverse propensity of treatment weighting for each patient, $q^{A=a}_\Phi = q(\Phi(X) \mid A = a)$ is the density function associated with the transformation $\Phi$ of the covariates $X$, the Integral Probability Metric (IPM) is a distance between distributions, and $\gamma$ is a positive constant.

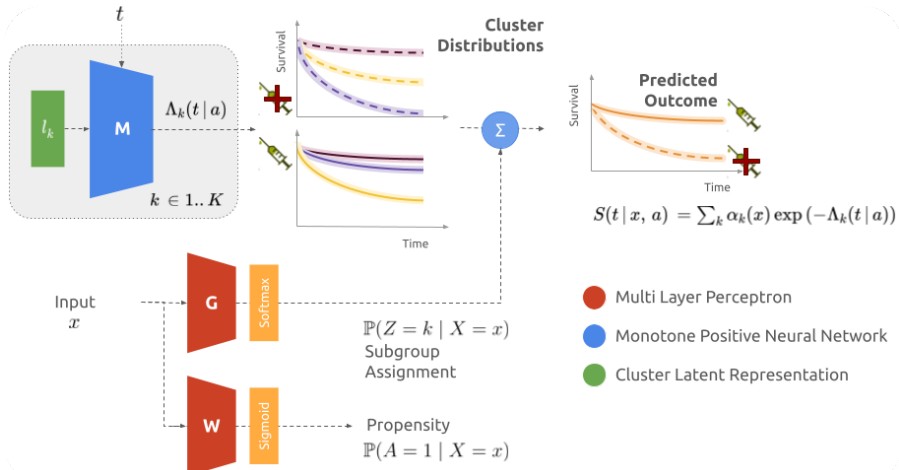

Figure 3: Causal Survival Clustering architecture. Latent parameter $l_k$ characterising the subgroup $k$ is inputted in the monotonic network $M$ to estimate the cumulative hazard $\Lambda_k$ under both treatment regimes. $G$ assigns the probability to belong to each subgroup given the patient's covariate(s) $x$. To tackle the challenge of treatment assignment bias, the network $W$ estimates the treatment propensity used to weigh the training likelihood. In this context, patient survival is the average of the weighted survival distributions across subgroups under the given treatment.

Shalit et al. propose to use this bound to train a neural network to estimate treatment effects. In this context, $\Phi$ is an inner representation of the network and the IPM regularisation renders this embedding similar between the treated and untreated populations. This embedding is then used to estimate the survival function under both treatment regimes. The difference between the estimated survival functions is then an accurate estimate of the treatment effect as the IPM penalisation corrects for the shift resulting from treatment non-randomisation.

Note that, under the dependencies we assume, only the subgroup assignment ($\mathbb{P}(Z \mid X)$) depends on the covariates $X$ and could, consequently, play the role of the transformation $\Phi$. However, treatment is assumed independent of the group membership under assumption 3, meaning that treatment rate does not differ between subgroups. This assumption results in the regularisation term being null, i.e. $\text{IPM}(q_{Z|X}^{A=0}, q_{Z|X}^{A=1}) = 0$ (see App. A.2 for derivation). This assumption is consequently necessary to correctly estimate the subgroups' treatment effect, without penalising the clustering structure.

As a consequence, our work focuses on the first term of the upper bound in equation 4 by re-weighting the factual likelihood with the patient's propensity score (Hassanpour & Greiner, 2019; Shalit et al., 2017). Using an estimator $\hat{p}_A(x) := \mathbb{P}(A = 1 \mid X = x)$ of the propensity score, we use a truncated propensity weighting (Austin & Stuart, 2015) scheme to avoid unstable weights in which the weights $w_i$ are defined as

$$w_i^{-1} = \begin{cases} 0.05 & \text{if } \hat{p}_A(x_i) < 0.05 \\ 0.95 & \text{if } \hat{p}_A(x_i) > 0.95 \\ a_i \cdot \hat{p}_A(x_i) + (1 - a_i) \cdot (1 - \hat{p}_A(x_i)) & \text{otherwise,} \end{cases} \tag{5}$$

where $i$ is the patient index, and $a_i$ is the realisation of $A$ for patient $i$. Using the factual likelihood from equation 2 and the weights $w_i$ defined in equation 5, we derive the upper-bound of the Negative Log-Likelihood (NLL), later used to train our model, as

$$\sum_{i, d_i = 1} w_i \log \left( -\frac{\partial S_\theta(u \mid x_i, a_i)}{\partial u} \bigg|_{u = t_i} \right) + \sum_{i, d_i = 0} w_i \log S_\theta(t_i \mid x_i, a_i) \tag{6}$$

## 2.2 ESTIMATING THE QUANTITIES OF INTEREST WITH NEURAL NETWORKS

The previous section discussed the quantities one must estimate —here parameterised by neural networks— to uncover subgroups of treatment effects: the assignment function $\mathbb{P}(Z \mid X)$, the

cumulative incidence $\Lambda_k$ characterising survival under the two treatment regimes, and the weights $w$. Figure 3 illustrates the overall architecture and the neural networks used to estimate these quantities.

**Subgroup assignment.** Similar to Jeanselme et al. (2022), a multi-layer perceptron $G$ with final Softmax layer assigns each patient characterised by covariates $x$ its probability of belonging to each subgroup, characterised through a $K$-dimensional vector of probabilities.

$$G(x) := [\mathbb{P}(Z = k \mid x)]_{k=1}^K$$

**Survival distributions.** Each subgroup $k$ is represented by a vector $l_k \in \mathbb{R}^L$ of dimension $L$, a latent parametrisation of the cumulative hazard functions. The vector $l_k$ is concatenated with $t$ and used as input to a neural network $M$ with monotonic positive outcomes[3], with a final SoftPlus layer to model the cumulative hazard functions under both treatment regimes $\Lambda_k(t) := (\Lambda_k(t \mid A = 0), \Lambda_k(t \mid A = 1))$. We use the following transformation to ensure that no probability is assigned to negative times, a limitation raised concerning previous monotonic neural networks (Shchur et al., 2020):

$$\Lambda_k(t) := t \cdot M(l_k, t)$$

By modelling these two cumulative intensity functions, we can estimate the Subgroup Average Treatment Effect, $\hat{\tau}_k(t)$, in subgroup $k$ as

$$\hat{\tau}_k(t) := \mathbb{E}[\tau(t, X) \mid Z = k)] = \exp(-\Lambda_k(t \mid A = 1)) - \exp(-\Lambda_k(t \mid A = 0))$$

**Inverse propensity weighting.** Under treatment randomisation, as in RCTs, the previous components would accurately model the observed survival outcomes and identify subgroups of treatment effects by maximising the factual likelihood. As previously discussed, to account for the treatment non-randomisation in observational studies, we weight the factual likelihood using the propensity score of a patient estimated through a multi-layer perceptron $W$ with a final sigmoid transformation as

$$W(x) := \mathbb{P}(A = 1 \mid x)$$

### 2.3 Training procedure

First, the network $W$ is trained to predict the binary treatment assignment by minimising the cross-entropy of receiving treatment. Then, training all other components relies on minimising the weighted factual log-likelihood introduced in equation 6. The use of monotonic neural networks results in the efficient and exact computation of the log-likelihood as automatic differentiation of the monotonic neural networks' outcomes readily provides the instantaneous hazards $\lambda_k$ necessary for computing the survival function derivative in the log likelihood (Jeanselme et al., 2022; 2023; Rindt et al., 2022).

## 3 Experimental analysis

As counterfactuals are unknown in observational data, we adopt—as is common practice in this research field—a synthetic dataset in which underlying survival distributions and group structure are known. Appendix C accompanies these synthetic results with the analysis of heterogeneity in real-world, adjuvant radiotherapy responses for patients diagnosed with breast cancer. Our code to produce the synthetic dataset, the model, and the reproduction of all experimental results is available on Github[4].

### 3.1 Data generation

We generate a population of $3,000$ equally divided into $K = 3$ subgroups. We draw 10 covariates from normal distributions with different centres and survival distributions for each treatment regime following Gompertz distributions (Pollard & Valkovics, 1992) parameterised by group membership and individual covariates. Note that this setting relaxes the independence of outcomes and

---

[3]To ensure this constraint, we apply a square function on all weights as proposed in Jeanselme et al. (2022; 2023).

[4]Link hidden for anonymity and code attached as supplementary material.

covariates assumed by the proposed model. This choice reflects a setting in which time-to-event distributions under treatment or control regimes are different functions of the covariates. This simulation is more likely to capture the complexity of real-world responses, in contrast to traditional evaluations of subgroup analysis which often assume a linear treatment response, i.e., a shift between treated and untreated distributions. Further, we implement two treatment assignment scenarios: a randomised assignment in which treatment is independent of patient covariates, similar to RCTs (**Randomised**), and one in which treatment is a function of the patient covariates as in observational studies (**Observational**). Finally, non-informative censoring times are drawn following a Gompertz distribution. Details of the generative process of our synthetic dataset are deferred to App. B.1.

## 3.2 EMPIRICAL SETTINGS

**Benchmark methods.** We compare the proposed approach against the state-of-the-art Cox Mixtures with Heterogeneous Effect (*CMHE* Nagpal et al. (2022a)[5]) which uncovers treatment effect and baseline survival latent groups. This method uses an expectation maximisation framework in which each patient is assigned to a group for which a Cox model is then fitted. The central differences with our proposed approach is that CMHE (i) clusters patients in treatment effects subgroups and survival subgroups, and (ii) assumes a linear treatment response. This separation between survival and treatment response improves the model's flexibility at the cost of interpretability as the number of groups grows exponentially, and subgroups of treatment effect are independent of survival. Further, the assumption of linear treatment response may hinder the discovery of subgroups with more complex responses. By contrast, our approach identifies subgroups of treatment while considering survival, without constraining the treatment response. We argue that these are key strengths of our method as a group not responding to treatment with low life expectancy would most benefit from alternative treatments, in comparison to a group with the same treatment responses but better survival odds. Considering jointly non-linear treatment effects and survival, therefore, results in identifying more clinically relevant subgroups.

For a fair comparison, we present three alternatives of CMHE: one with fixed $K = 3$ survival subgroups, one with $L = 3$ treatment effect subgroups, and one with $K = L = 2$, which allows for a total of 4 subgroups. Crucially, these methods assume proportional hazards for each subgroup and do not account for the treatment non-randomisation. Additionally, we also compare our model against its unadjusted alternative (CSC Unadjusted), which uses the unweighted factual likelihood ($w_i = 1$) and should suffer from non-randomisation of treatment assignment.

Finally, we compare against two step-wise approaches in which clustering and treatment effect are modelled separately. First, we use an unsupervised clustering algorithm on the covariates, followed by a non-parametric estimate of the treatment effect as proposed in Nagpal et al. (2022a), referred as *Kmeans + TE* in the following. Specifically, we use a K-Means (Hartigan & Wong, 1979) to cluster the data and we compute the difference between Kaplan-Meier (Kaplan & Meier, 1958) estimates between control and treated patients stratified by clusters. However, this approach is limited by its implicit assumption of randomisation and its pre-specified number of subgroups, which is not informed by observed outcomes. Second, we use a *Virtual Twins* approach to estimate individualised treatment effects and then cluster them. As proposed in the literature, we propose to use a survival tree to estimate response under each treatment regime and then use a KMeans on the estimated treatment effects, computed as the difference between survival estimates. Note that this methodology also assumes treatment randomisation, and the clustering is uninformed by the covariates. For details on training and hyperparameter optimisation, we refer to App. B.2.

**Evaluation.** In the synthetic experiments, the subgroup structure is known. We measure the adjusted[6] Rand-Index (Rand, 1971), which quantifies how the estimated assignment aligns with the known underlying group structure. Additionally, we use the integrated absolute error (IAE) between the treatment effect estimate and the ground truth, which measures how well we recover each subgroup's treatment effect

$$\text{IAE}_k(t) = \int_0^t \left| \hat{\tau}_{\hat{k}}(s) - \tau_k(s) \right| ds,$$

---

[5]Implemented in the Auton-Survival library (Nagpal et al., 2022b)

[6]Random patient assignment results in an adjusted Rand-Index of 0.

where $\hat{\tau}_{\hat{k}}$ is the estimated treatment effect for subgroup $\hat{k} = \arg\max_l \mathbb{E}_{x \in k}(Z = l \mid x)$, i.e. the most likely assigned cluster for patients in the underlying $k$ cluster, and $\tau_k$ is the ground truth.

### 3.3 TREATMENT EFFECT RECOVERY

In this section, we present the results under the previously described data generation process. App. B.4 presents alternative datasets which demonstrate the flexibility of the proposed methodology.

**Recovering the underlying number of subgroups.** For all methodologies, we need to choose the number of subgroups a priori. An important question is, therefore, whether we can identify the true number of subgroups with our model. Figure 4 presents the average NLL obtained by cross-validation for models with different numbers of subgroups $K$. The dotted lines represent the elbow heuristic (Thorndike, 1953), which identifies a change point in the explained variability of a clustering strategy, here considering the log-likelihood. Using this heuristic, the optimal choice for $K$ is 3, which agrees with the underlying generative process. App. B.3 explores the sensitivity to this choice, demonstrating the method's robustness. This data-driven choice of the number of subgroups $K$ is a crucial strength of our method, compared to classical two-stage analyses which separate clustering from treatment effect estimates. In these methods, survival outcomes cannot directly guide the choice of $K$.

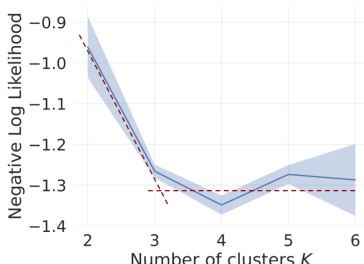

Figure 4: Averaged negative log-likelihood across 5-fold cross-validation given the number of subgroups $K$ under the "Observational" treatment assignment with the shaded area representing 95% CI. The log-likelihood presents an elbow around the underlying number of subgroups.

**Discovering subgroups.** Next, we present quantitative results of our method against the benchmarks methods. Table 1 presents the performance of the different methodologies under the two studied scenarios. Recall that $K$ denotes the number of treatment response subgroups, while in CMHE, the additional parameter $L$ describes the number of survival distributions.

| | Model | Rand-Index | $k=1$ | IAE$_k(t_{max})$ $k=2$ | $k=3$ |
|---|---|---|---|---|---|
| **Randomised** | **CSC** | *0.798* (0.054) | 0.042 (0.012) | 0.032 (0.007) | **0.015** (0.006) |
| | **CSC Unadjusted** | **0.804** (0.052) | *0.032* (0.009) | **0.027** (0.007) | *0.018* (0.010) |
| | CMHE ($K=3$) | 0.392 (0.034) | 0.164 (0.009) | 0.077 (0.004) | 0.070 (0.005) |
| | CMHE ($L=3$) | 0.258 (0.155) | - | 0.088 (0.007) | - |
| | CMHE ($K=L$) | 0.392 (0.054) | 0.338 (0.262) | - | - |
| | KMeans + TE | 0.000 (0.002) | 0.210 (0.011) | 0.091 (0.007) | 0.142 (0.016) |
| | Virtual Twins | 0.578 (0.081) | **0.022** (0.006) | 0.032 (0.011) | 0.025 (0.010) |
| **Observational** | **CSC** | **0.812** (0.038) | **0.037** (0.013) | *0.034* (0.009) | **0.022** (0.005) |
| | **CSC Unadjusted** | *0.727* (0.084) | *0.048* (0.019) | **0.025** (0.011) | *0.025* (0.009) |
| | CMHE ($K=3$) | 0.385 (0.022) | 0.169 (0.012) | 0.078 (0.005) | 0.075 (0.008) |
| | CMHE ($L=3$) | 0.190 (0.127) | 0.192 (0.010) | - | 0.140 (0.009) |
| | CMHE ($K=L$) | 0.454 (0.068) | 0.210 (0.013) | 0.095 (0.020) | 0.188 (0.014) |
| | KMeans + TE | 0.001 (0.004) | 0.192 (0.016) | 0.090 (0.019) | 0.147 (0.016) |
| | Virtual Twins | 0.438 (0.139) | 0.050 (0.039) | *0.034* (0.033) | 0.051 (0.033) |

Table 1: 5-fold cross-validated performance averaged (with standard deviation in parenthesis) under the Randomised and Observational treatment simulation. Best performance per column and simulation scenario are marked in **bold**, second best in *italic*. '-' describes when the methodology diverges. Our proposed CSC method and its unadjusted variant best recovers the underlying treatment responses, with the adjusted approach presenting the best performance in the observational setting.

*CSC outperforms all CMHE alternative, the current state-of-the-art method.* CMHE's parametrisation and assumptions explain this difference. Critically, CMHE assumes (i) proportional hazards and (ii) a linear impact of treatment on log-hazards. Neither of these two assumptions is likely to hold in

real-world settings as mimicked by our synthetic data. In contrast, our approach does not constrain the treatment effect due to its flexible modelling of the survival function under both treatment regimes. Increasing the number of subgroups, as shown in $K = L$ improves the performance of CMHE in terms of clustering quality (Rand-Index) and the recovery of the underlying treatment effect (lower IAE), but still is inferior compared to our proposed methods. These experiments highlight the advantage of the proposed CSC method in uncovering subgroups of treatment responses due to the flexibility in modelling complex survival distributions under both treatment regimes without proportionality assumption.

*CSC presents the best performance in identifying the underlying subgroup.* Our proposed CSC method outperforms all approaches on the Rand-Index. In particular, the two-step approaches do not identify the underlying subgroups when covariates and outcomes subgroups are aligned. KMeans + TE identify subgroups that are independent of the outcome of interest, as shown by the Rand-Index. Consequently, the identified clusters differ in their treatment responses as shown by the large IAE. The Virtual Twins approach presents a better capacity to identify the clusters of interest with better Rand-Index and IAE. However, the two-step approach hurts subgroup identification due to the disconnect between the clustering and input covariates. This disconnect leads to a significantly lower Rand-Index in comparison to CSC despite accurate treatment effect estimates.

*Treatment assignment correction improves subgroup identification in observational settings.* The Observational simulation demonstrates the importance of correcting the likelihood under treatment non-randomisation. The two CSC alternatives present comparable performance in the randomised setting, as theoretically expected, due to a constant $w_i$ in this context. Note that estimating propensity slightly reduces performance due to error introduced, reducing performance for the corrected alternative. In a randomised setting, CSC Unadjusted should be preferred. However, the superiority of CSC over its unadjusted alternative is evident in the observational setting, where CSC better recovers the different groups as shown by the Rand-Index and subgroup treatment effects. Finally, Appendices B.4.1 to B.4.5 further validate the robustness of our method in settings with unequally distributed cluster sizes, population size, increased number of underlying clusters, different treatment rates and covariate structures.

**Real-world heterogeneity analysis in adjuvant radiotherapy responses (App. C).** Using the Surveillance, Epidemiology, and End Results program[7](SEER), App. C investigates a real-world setting: we analyse which subgroups of breast cancer patients most benefit from adjuvant radiotherapy, an active area of research. Our analysis highlights a subgroup characterised by a higher number of distant lymph nodes that benefit from treatment. Despite the inherent limitations of the dataset, our exploratory method identifies a subgroup that alternative methods miss, potentially informing future clinical trials to validate or challenge this hypothesis. Critically, this demonstrates that our work offers a practical methodology for exploratory subgroup analysis of medical data to identify treatment effect heterogeneity in observational studies.

## 4 RELATED WORK

While survival subgrouping has been proposed through mixtures of distributions (Nagpal et al., 2021a;b; Jeanselme et al., 2022) to identify different phenotypes of patients, the machine learning literature on phenotyping treatment effects with time-to-event outcomes remains sparse. The current literature focuses on estimating population or individual treatment effects (Curth et al., 2021; Shalit et al., 2017; Johansson et al., 2016; Zhang et al., 2020). While these models help to understand the average population-wide response to treatment and aim to estimate individualised treatment effects, they do not provide an understanding of the patient groups that may benefit more or be harmed more by treatment. Identifying such groups aligns with, and is thus more useful for, drafting medical guidelines that direct treatment to those subgroups most likely to benefit from it.

Discovering intervenable subgroups is core to medical practice, particularly identifying subgroups of treatment effect, as patients do not respond like the average (Bica et al., 2021; Ruberg et al., 2010; Sanchez et al., 2022) and the average may conceal differential treatment responses. Identification of subgroups has long been used to design RCTs. Indeed, subgroups identified *a priori* can then be tested through trials (Cook et al., 2004; Rothwell, 2005). *A posteriori* analyses have gained traction

---

[7]Available at `https://seer.cancer.gov/`

to uncover subgroups of patients from existing RCTs to understand the underlying variability of responses.

The first set of subgroup analysis methodologies consists of a step-wise approach: (i) estimate the ITE and (ii) uncover subgroups using a second model to explain the heterogeneity in ITE. Foster et al. (2011); Qi et al. (2021) describe the virtual twins approach in which one models the outcome using a decision tree for each treatment group. The difference between these decision trees results in the estimated treatment effects. A final decision tree aims to explain these estimated treatment effects to uncover subgroups. Similar approaches have been explored with different meta-learners (Xu et al., 2023), or Bayesian additive models (Hu et al., 2021), or replacing the final step with a linear predictor to uncover the feature influencing heterogeneity (Chernozhukov et al., 2018). However, Guelman et al. (2015) discuss the drawbacks associated with these approaches. Notably, the two-step optimisation may not lead to recovery of the underlying subgroups of treatment effects.

Tree-based approaches were proposed to address the limitations of step-wise approaches by jointly discovering subgroups and modelling the treatment effect. Instead of traditional splits on the observed outcomes, these causal trees aim to discover homogeneous splits regarding covariates and treatment effects. Su et al. (2009) introduce a recursive population splitting based on the average difference in treatment effect between splits. Athey & Imbens (2015; 2016) improve the confidence interval estimation through the honest splitting criterion, which dissociates the splitting from the treatment effect estimation. Wager & Athey (2018) agglomerate these causal trees into causal forests for improved ITE estimation. Each obtained split in the decision tree delineates two subgroups of treatment effect Lipkovich et al. (2011); Loh et al. (2015). Alternatively, McFowland III et al. (2018) propose pattern detection and Wang & Rudin (2022), causal rule set learning to uncover these subgroups. However, all these approaches rely on a local optimisation criterion (Lipkovich & Dmitrienko, 2014) and greedy split exploration. Recently, Nagpal et al. (2020) addressed the local optimisation by constraining the treatment response to a linear form in a mixture of Cox models.

Previous approaches uncover subgroups of treatment effect but consider *RCTs with binary outcomes*, not the observational setting with inherently continuous survival outcomes that our paper explores. At the intersection with survival analysis, Zhang et al. (2017) extend causal trees to survival causal trees, modifying the splitting criterion by measuring the difference in survival estimates between resulting leaves. Similarly, Hu et al. (2021) propose Bayesian additive models and Zhu & Gallego (2020) propose a step-wise approach with propensity weighting to study observational data. Closest to our work, (Jia et al., 2021; Nagpal et al., 2023) propose to uncover subgroups within RCTs with survival outcomes. Jia et al. (2021) propose a mixture of treatment effects characterised by Weibull distributions trained in an expectation-maximisation framework. Similarly, Nagpal et al. (2023) stratify the population into three groups: non-, positive- and negative responders to treatment. An iterative Monte Carlo optimisation is used to uncover these subgroups, characterised by a Cox model with a multiplicative treatment effect. As demonstrated in our work, this step-wise optimisation may be limiting, and the assumption of RCTs renders the model less relevant in observational data.

## 5 CONCLUSION

This work fills the current gap in the literature of methodologies to identify subgroups of treatment responses in observational time-to-event data. While modelling the observed outcome, the approach jointly identifies patient subgroups with different treatment responses. Our experiments demonstrate the capacity of our proposed method to uncover groups of treatment effects in both synthetic and real-world datasets. However, causal modelling relies on empirically unverifiable assumptions. The invalidity of these assumptions is a lesser concern in our targeted setting of hypothesis generation, in comparison to individual estimation of treatment recommendations.

Serving the purpose of a hypothesis-generating tool, we invite practitioners to further investigate the subgroups identified in observational studies through RCTs to (i) validate the estimated responses, (ii) identify potential alternative treatments with improved responses, and (iii) inform clinical guidelines.

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

# A PROOF

## A.1 INDIVIDUALISED TREATMENT EFFECT

This section derives the individualised treatment effect expression introduced in equation 3.

$$
\begin{aligned}
\tau(t, x) :&= \mathbb{E}(\mathbb{1}(T_1 \geq t) - \mathbb{1}(T_0 \geq t) \mid X = x) \\
&= \mathbb{E}(\mathbb{1}(T_1 \geq t) \mid X = x) - \mathbb{E}(\mathbb{1}(T_0 \geq t) \mid X = x) \\
&= \mathbb{P}(T' \geq t \mid A = 1, X = x) - \mathbb{P}(T' \geq t \mid A = 0, X = x) \quad \text{(Under assumption 4 and 5)}
\end{aligned}
$$

## A.2 UPPER-BOUND SIMPLIFICATION

In Section 2, we claim that under the considered DAG presented in Figure 2,

$$
\text{IPM}(q_{Z|X}^{A=0}, q_{Z|X}^{A=1}) = 0
$$

with $A$, being the treatment assignment. Under assumption 3, expressing this quality corresponds to:

$$
q_{Z|X}^{A=a} := q(Z \mid X, A = a) = q(Z \mid X)
$$

with $q$, the density function. From this expression, $q_{Z|X}^{A=0} = q_{Z|X}^{A=1}$, which results in the distance between these distributions being null.

# B SYNTHETIC ANALYSIS

## B.1 DATA GENERATION

We consider a synthetic population of $N = 3,000$ patients with 10 associated covariates $X \in \mathbb{R}^{10}$ divided into $K = 3$ subgroups. The following data generation does not aim to mimic a particular real-world setting but follows a similar approach to Nagpal et al. (2022a). The following describes our generation process:

**Covariates.** Each patient's membership $Z$ is drawn from a multinomial with equal probability. Group membership informs the two first covariates through the parametrisation of the bivariate normal distribution with centres $c_k$ equal to $(0, 2.25)$, $(-2.25, -1)$, and $(2.25, -1)$. All other covariates are drawn from standard normal distributions. Formally, this procedure is described as:

$$Z \sim \text{Mult}\left(1, \left[\frac{1}{3}, \frac{1}{3}, \frac{1}{3}\right]\right)$$

$$X_{[1,2]} \mid Z = k \sim \text{MVN}(c_k, I^2)$$

$$Y \sim \text{Mult}\left(1, \left[\frac{1}{4}, \frac{1}{4}, \frac{1}{4}, \frac{1}{4}\right]\right)$$

$$X_{[3:10]} \mid Y = k \sim \text{MVN}(c'_k, I^8)$$

with MVN denoting a multivariate normal distribution, $c'_k$ random cluster centres, and $I^n$, the identity covariance matrix of dimension $n$. Note that we introduce $Y$ to present a covariates structure that is independent of the treatment responses.

**Treatment response.** For each subgroup, event times under treatment and control regimes are drawn from Gompertz distributions, with parameters that are functions of group-specific coefficients ($B^0$ and $\Gamma^0$ for the event time when untreated and $B^1$ and $\Gamma^1$ when treated) and the patient's covariates.

$$B^0_z \mid Z = z \sim \text{MVN}(0, I^{10})$$

$$\Gamma^0_z \mid Z = z \sim \text{MVN}(0, I^{10})$$

$$T_0 \mid Z, X, B^0_z, \Gamma^0_z = (z, x, \beta^0_z, \gamma^0_z) \sim \text{Gompertz}\left(w_0(\beta^0_z, x), s_0(\gamma^0_z, x)\right)$$

$$B^1_z \mid Z = z \sim \text{MVN}(0, I^{10})$$

$$\Gamma^1_z \mid Z = z \sim \text{MVN}(0, I^{10})$$

$$T_1 \mid Z, X, B^1_z, \Gamma^1_z = (z, x, \beta^1_z, \gamma^1_z) \sim \text{Gompertz}\left(w_1(\beta^1_z, x), s_1(\gamma^1_z, x)\right)$$

with $w_0$, $w_1$ two functions parametrising the Gompertz distributions' shape as $w_0(\beta, x) := |\beta[0]| + (x[5 : 10] \cdot \beta[5 : 10])^2$, $w_1(\beta, x) := |\beta[0]| + (x[1 : 5] \cdot \beta[1 : 5])^2$, and the shift parameter parameterised as $s_0(\gamma, x) := |\gamma[0]| + |(x[1 : 5] \cdot \gamma[1 : 5])|$ and $s_1(\gamma, x) := |\gamma[0]| + |(x[5 : 10] \cdot \gamma[5 : 10])|$ where $v[i]$ described the $i^{th}$ element of the vector $v$. These functions aim to introduce non-linear responses with discrepancies between control and treatment regimes. Note that we allow covariates to influence the survival distribution as a patient's covariates influence Gompertz's shapes and scales.

**Treatment assignment.** The non-randomisation of treatment is central to the problem of identifying treatment subgroups in real-world applications. Assuming a treatment assignment probability of 50%, we assign each patient to a given treatment. We propose two treatment assignment strategies reflecting a RCT and an observational setting, denoted as "Randomised" and "Observational". "Randomised" consists of a Bernoulli draw using the realisation of $P$. "Observational" reflects an assignment dependent upon the observed covariates.

$$A_{rand} \mid P = p \sim \text{Bernoulli}(0.5)$$

$$A_{obs} \mid P, X = (p, x) \sim \text{Bernoulli}(F_{\Phi(X)}(\Phi(x)) \times 0.5)$$

with $F_{\Phi(X)}(\Phi(x))$ the cumulative distribution function that returns the probability that a realisation of a $\Phi(X)$ will take a smaller value than $\Phi(x)$. In our experiment, we chose $\Phi(x) = \sum x^2$.

**Censoring.**     Finally, our work focuses on right-censored data. To generate censoring independent of the treatment and event, we draw censoring from another Gompertz distribution as follows:

$$B^C \sim \text{MVN}(0, I^5)$$
$$C \mid X, B^C = (x, \beta) \sim \text{Gompertz}\left(w_c(\beta, x), 0\right)$$
$$T' = A \cdot T_1 + (1 - A) \cdot T_0$$
$$T = \min(C, T')$$
$$D = \mathbb{1}(C > T')$$

with $w_c := (x[5:10] \cdot \beta)^2$, the scale of the censoring Gompertz distribution.

From these data, the goal is to model the treatment effect from the observed $X, T, D, A$ with $T \mid A = a, T_0, T_1, C = t_0, t_1, c := \min(c, (1-a) * t_0 + a * t_1)$ and $D = \mathbb{1}_{C>T}$.

## B.2   TRAINING AND HYPERPARAMETER OPTIMISATION

**Training.**     We perform a 5-fold cross-validation for both Randomised and Observational simulations. For each cross-validation split, the development set is divided into three parts: 80% for training, 10% for early stopping, and 10% for hyper-parameter search over a grid presented in Appendix B.2. All models were optimised for 1000 epochs using an Adam optimiser (Kingma & Ba, 2015) with early stopping.

**Hyperparameter optimisation.**     We adopted a 100-iteration random grid-search over the following hyperparameters: network depth between 1 and 3 inner layers with 50 nodes, latent subgroup representation in $[10, 25, 50]$ and for training, a learning rate of $0.001$ or $0.0001$ with batch size of $100$ or $250$.

## B.3   SENSITIVITY

In Section 3, we describe how the proposed methodology presents an elbow in the likelihood as a function of the number of clusters around the underlying number of clusters $K = 3$. This section explores how the identified treatment effects change under a misspecified model which estimates 2 and 4 clusters. This analysis examines the risk associated with misspecifying the number of clusters.

Figure 5 illustrates the cross-validated subgroup treatment effects when trained with $K = 2, 3$ and 4 clusters. Uncertainty increases with misspecified models due to the instability of the identified subgroups.

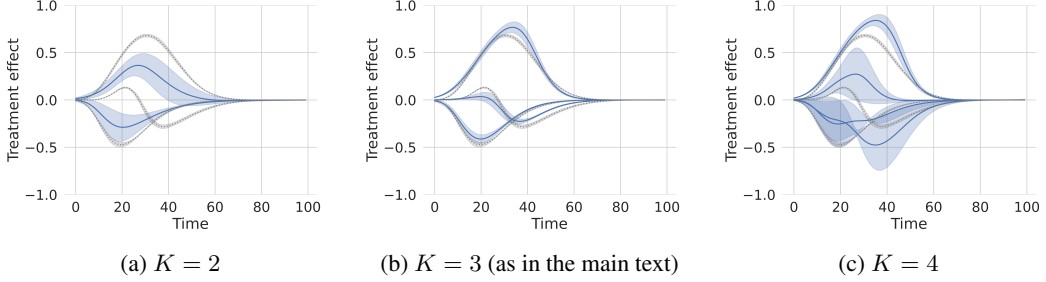

(a) $K = 2$          (b) $K = 3$ (as in the main text)          (c) $K = 4$

Figure 5: Sensitivity to the number of clusters used to train CSC. Lines in blue represent the cross validated average estimated treatment effect. Lines in grey corresponds to the ground truth.

## B.4 Alternative data generations

In this section, we explore alternative data generations to demonstrate the robustness of the proposed strategy.

### B.4.1 Unequal cluster size

Section B.1 presents a population equally distributed over the different clusters. In medical settings, the underlying subgroups may differ in size. This section explores an alternative scenario in which the population is distributed over the three clusters: 62.5%, 25% and 12.5%.

Table 2 summarises the performance in this simulation, echoing the main text's conclusions. The proposed method best recovers the different subgroups and presents one of the best estimates of subgroups' treatment effects..

| | Model | Rand-Index | $\text{IAE}_k(t_{max})$ $k=1$ | $k=2$ | $k=3$ |
|---|---|---|---|---|---|
| **Randomised** | **CSC** | *0.637* (0.068) | 0.064 (0.013) | *0.021* (0.009) | **0.062** (0.010) |
| | **CSC Unadjusted** | **0.651** (0.049) | *0.055* (0.009) | **0.017** (0.010) | *0.067* (0.010) |
| | CMHE ($K=3$) | 0.589 (0.019) | 0.179 (0.011) | 0.096 (0.008) | 0.086 (0.004) |
| | CMHE ($L=3$) | 0.072 (0.051) | 0.208 (0.004) | 0.211 (0.027) | 0.122 (0.011) |
| | CMHE ($K=L$) | 0.471 (0.022) | 0.241 (0.011) | 0.264 (0.016) | 0.173 (0.010) |
| | KMeans + TE | 0.336 (0.010) | 0.078 (0.008) | 0.029 (0.007) | 0.237 (0.017) |
| | Virtual Twins | 0.541 (0.163) | **0.033** (0.025) | 0.052 (0.018) | 0.087 (0.051) |
| **Observational** | **CSC** | **0.780** (0.086) | 0.064 (0.027) | *0.040* (0.011) | **0.069** (0.010) |
| | **CSC Unadjusted** | *0.616* (0.146) | 0.072 (0.020) | **0.039** (0.028) | 0.117 (0.033) |
| | CMHE ($K=3$) | 0.611 (0.022) | 0.179 (0.014) | 0.103 (0.005) | 0.089 (0.005) |
| | CMHE ($L=3$) | 0.084 (0.034) | 0.211 (0.007) | 0.209 (0.011) | 0.127 (0.009) |
| | CMHE ($K=L$) | 0.462 (0.062) | 0.252 (0.012) | 0.286 (0.008) | 0.096 (0.010) |
| | KMeans + TE | 0.341 (0.020) | *0.045* (0.009) | 0.043 (0.009) | 0.275 (0.017) |
| | Virtual Twins | 0.534 (0.166) | **0.040** (0.015) | 0.051 (0.024) | *0.084* (0.035) |

Table 2: Cross-validated performance (with standard deviation in parenthesis) when clusters are of different sizes.

### B.4.2 Number of Points

Section B.1 describes a data generation with $3,000$ patients. This section presents results when there are 300 and $30,000$ patients. A smaller number of patients may result in a larger impact of treatment non-randomisation but also impact the neural network capacity to identify underlying subgroups of treatment effect. A larger number makes non-randomisation less of a concern.

Table 3 presents the performance with these different population sizes under an observational treatment assignment. A first observation is that all methodologies present better performance with larger number of points. Further, baselines that do not account for the assignment mechanisms present lower performance with $N = 300$ as non-randomisation has an increased impact on treatment effect estimate with a smaller population. This difference decreases when $N = 30,000$ with the virtual twins approach presenting the best recovery of the treatment effects. However, throughout the different settings, the proposed CSC presents the best Rand-Index indicating a good recovery of the underlying structure.

| | Model | Rand-Index | $k = 1$ | $\text{IAE}_k(t_{max})$
$k = 2$ | $k = 3$ |
|---|---|---|---|---|---|
| **300** | **CSC** | **0.504** (0.139) | **0.131** (0.090) | **0.085** (0.033) | **0.091** (0.062) |
| | **CSC Unadjusted** | *0.449* (0.106) | 0.202 (0.049) | 0.162 (0.055) | 0.125 (0.088) |
| | CMHE ($K = 3$) | 0.126 (0.084) | 0.340 (0.050) | 0.135 (0.043) | 0.166 (0.050) |
| | CMHE ($L = 3$) | 0.131 (0.067) | - | - | 0.254 (0.022) |
| | CMHE ($K = L$) | 0.087 (0.118) | 0.284 (0.056) | 0.639 (0.939) | 0.747 (1.014) |
| | KMeans + TE | 0.050 (0.110) | 0.251 (0.072) | *0.154* (0.065) | 0.267 (0.049) |
| | Virtual Twins | 0.296 (0.115) | *0.159* (0.072) | 0.172 (0.067) | *0.101* (0.043) |
| **30,000** | **CSC** | **0.774** (0.015) | *0.018* (0.010) | 0.012 (0.003) | *0.030* (0.009) |
| | **CSC Unadjusted** | *0.737* (0.022) | 0.033 (0.006) | *0.010* (0.002) | 0.039 (0.008) |
| | CMHE ($K = 3$) | 0.567 (0.159) | 0.094 (0.011) | 0.059 (0.018) | 0.142 (0.034) |
| | CMHE ($L = 3$) | 0.622 (0.040) | 0.078 (0.001) | 0.125 (0.003) | 0.226 (0.003) |
| | CMHE ($K = L$) | 0.638 (0.008) | 0.059 (0.001) | 0.220 (0.007) | 0.140 (0.007) |
| | KMeans + TE | 0.000 (0.000) | 0.087 (0.004) | 0.147 (0.008) | 0.208 (0.004) |
| | Virtual Twins | 0.650 (0.030) | **0.011** (0.003) | **0.009** (0.000) | **0.010** (0.004) |

Table 3: Cross-validated performance (with standard deviation in parenthesis) with varying $N$ under observational treatment setting.

### B.4.3 NUMBER OF CLUSTERS

Section B.1 describes a data generation with $K = 3$ clusters. This section explores the behaviour of the different methodologies with an increased number of clusters $K = 5$. Following the same data generation, we introduces two additional clusters with centres [-3, 3] and [4, 4].

Table 4 presents the different models' performance in this setting, echoing the same results as the main text. The proposed method best recovers the underlying clustering structure as shown by the Rand-Index. CMHE suffers from its inherent association between treatment and survival. The step-wise approach is unable to identify the clusters of interest if the data presents a structure that is not fully aligned with the outcome of interest.

This scenario further validates our findings, demonstrating the methodology's capacity to generalise beyond the scenario presented in the main text.

| | Model | Rand-Index | $k = 1$ | $k = 2$ | $\mathrm{IAE}_k(t_{max})$ $k = 3$ | $k = 4$ | $k = 5$ |
|---|---|---|---|---|---|---|---|
| Randomised | **CSC** | **0.475** (0.038) | *0.073* (0.030) | *0.059* (0.023) | 0.041 (0.015) | 0.099 (0.024) | **0.017** (0.003) |
| | **CSC Unadjusted** | *0.459* (0.037) | 0.095 (0.055) | **0.054** (0.027) | 0.046 (0.011) | 0.076 (0.020) | *0.019* (0.005) |
| | CMHE ($K = 5$) | 0.306 (0.016) | 0.091 (0.036) | 0.104 (0.012) | **0.015** (0.002) | *0.093* (0.011) | 0.232 (0.013) |
| | CMHE ($L = 5$) | 0.256 (0.018) | 0.135 (0.020) | 0.112 (0.008) | *0.020* (0.003) | 0.221 (0.009) | 0.349 (0.006) |
| | CMHE ($K = L$) | 0.372 (0.018) | 0.101 (0.023) | 0.181 (0.013) | 0.072 (0.003) | 0.119 (0.010) | 0.323 (0.010) |
| | KMeans + TE | 0.156 (0.012) | 0.075 (0.009) | 0.184 (0.016) | 0.072 (0.012) | 0.147 (0.014) | 0.039 (0.010) |
| | Virtual Twins | 0.284 (0.081) | **0.034** (0.029) | 0.066 (0.026) | 0.029 (0.012) | **0.082** (0.023) | 0.036 (0.010) |
| Observational | **CSC** | **0.446** (0.016) | 0.119 (0.072) | 0.087 (0.018) | **0.026** (0.008) | 0.097 (0.023) | **0.027** (0.016) |
| | **CSC Unadjusted** | *0.444* (0.048) | *0.077* (0.046) | 0.065 (0.026) | *0.045* (0.020) | 0.118 (0.026) | 0.034 (0.022) |
| | CMHE ($K = 5$) | 0.303 (0.010) | 0.095 (0.030) | 0.112 (0.017) | 0.018 (0.005) | *0.094* (0.013) | 0.229 (0.010) |
| | CMHE ($L = 5$) | 0.217 (0.026) | 1.341 (2.717) | 0.119 (0.014) | 0.022 (0.005) | 1.432 (2.720) | 1.561 (2.709) |
| | CMHE ($K = L$) | 0.371 (0.029) | 0.105 (0.009) | 0.208 (0.043) | 0.076 (0.006) | 0.186 (0.025) | 0.325 (0.004) |
| | KMeans + TE | 0.162 (0.011) | **0.058** (0.005) | 0.210 (0.031) | 0.084 (0.018) | 0.122 (0.032) | *0.032* (0.006) |
| | Virtual Twins | 0.320 (0.084) | *0.062* (0.038) | **0.026** (0.008) | 0.044 (0.021) | **0.077** (0.040) | 0.038 (0.018) |

Table 4: Cross-validated performance (with standard deviation in parenthesis) when $K = 5$.

### B.4.4 IMPACT OF TREATMENT RATE

Section B.1 assumes 50% of the population receives treatment. This section explores when 25% and 75% of the population receive treatment under the non-randomised treatment setting.

Table 5 presents the performance under these different treatment rates in an observational setting. These results highlight CSC's capacity to identify the underlying subgroups with the highest Rand-Index and one of the best cluster treatment effect recovery in these settings.

| | Model | Rand-Index | $IAE_k(t_{max})$ $k = 1$ | $k = 2$ | $k = 3$ |
|---|---|---|---|---|---|
| 25% | **CSC** | *0.790* (0.036) | *0.036* (0.009) | *0.034* (0.009) | *0.018* (0.006) |
| | **CSC Unadjusted** | **0.849** (0.033) | 0.040 (0.005) | **0.019** (0.005) | **0.014** (0.004) |
| | CMHE ($K = 3$) | 0.376 (0.028) | 0.168 (0.007) | 0.077 (0.002) | 0.073 (0.004) |
| | CMHE ($L = 3$) | 0.149 (0.080) | 0.188 (0.004) | - | 0.154 (0.005) |
| | CMHE ($K = L$) | 0.410 (0.038) | 0.223 (0.011) | - | - |
| | KMeans + TE | 0.001 (0.006) | 0.187 (0.005) | 0.088 (0.014) | 0.149 (0.011) |
| | Virtual Twins | 0.527 (0.060) | **0.034 (0.012)** | **0.019** (0.005) | 0.034 (0.012) |
| 75% | **CSC** | **0.808** (0.043) | *0.035* (0.012) | *0.027* (0.006) | **0.019** (0.007) |
| | **CSC Unadjusted** | *0.687* (0.048) | 0.045 (0.011) | 0.035 (0.016) | 0.028 (0.006) |
| | CMHE ($K = 3$) | 0.385 (0.039) | 0.168 (0.010) | 0.077 (0.005) | 0.069 (0.004) |
| | CMHE ($L = 3$) | 0.318 (0.140) | - | - | 0.132 (0.008) |
| | CMHE ($K = L$) | 0.483 (0.104) | 0.208 (0.013) | - | - |
| | KMeans + TE | 0.002 (0.003) | 0.195 (0.015) | 0.079 (0.012) | 0.148 (0.011) |
| | Virtual Twins | 0.570 (0.013) | **0.030** (0.005) | **0.024** (0.013) | *0.022* (0.005) |

Table 5: Cross-validated performance (with standard deviation in parenthesis) with varying treatment rates.

### B.4.5 ALIGNED COVARIATES AND TREATMENT EFFECT STRUCTURE

Section B.1 presents a data generation with the last 8 covariates having a clustered structures independent of the treatment response. This section explores a setting where all clusters in the covariates are associated with different treatment responses of interest. Specifically, we sample the last dimensions from standard normal distributions instead of various clusters, i.e.,

$$X_{[3-10]} \sim \text{MVN}(0, I^8)$$

Table 6 summarises the performance in this setting, evidencing an improvement of the Kmeans+TE approach. When covariates' clusters are aligned with the outcome of interest, this method recovers the clustering structure well, as shown by the best Rand-Index. Non-randomisation of treatment, however, worsens the treatment estimates due to the absence of correction in the estimation of treatment responses.

Even in this unrealistic scenario, our proposed method remains the second best in recovering the underlying clustering structure and associated treatment responses. This observation reinforces the main findings of our work, demonstrating the method's potential to identify subgroups of treatment effects across settings.

| | Model | Rand-Index | IAE$_k(t_{max})$ | | |
| | | | $k = 1$ | $k = 2$ | $k = 3$ |
|---|---|---|---|---|---|
| Randomised | **CSC** | 0.487 (0.052) | 0.027 (0.013) | 0.047 (0.020) | *0.030* (0.009) |
| | **CSC Unadjusted** | *0.522* (0.041) | *0.025* (0.004) | *0.027* (0.018) | 0.034 (0.013) |
| | CMHE ($K = 3$) | 0.425 (0.020) | 0.059 (0.004) | 0.034 (0.025) | 0.134 (0.012) |
| | CMHE ($L = 3$) | 0.165 (0.107) | 0.062 (0.004) | 0.126 (0.175) | 0.233 (0.179) |
| | CMHE ($K = L$) | 0.289 (0.078) | 0.063 (0.009) | 0.050 (0.019) | 0.146 (0.020) |
| | KMeans + TE | **0.888** (0.020) | **0.015** (0.007) | **0.021** (0.007) | **0.020** (0.003) |
| | Virtual Twins | 0.200 (0.090) | 0.035 (0.011) | 0.065 (0.025) | 0.057 (0.030) |
| Observational | **CSC** | *0.666* (0.146) | *0.016* (0.007) | **0.029** (0.013) | 0.033 (0.010) |
| | **CSC Unadjusted** | 0.613 (0.207) | 0.024 (0.016) | 0.039 (0.015) | *0.030* (0.003) |
| | CMHE ($K = 3$) | 0.392 (0.062) | 0.059 (0.008) | *0.034* (0.019) | 0.131 (0.015) |
| | CMHE ($L = 3$) | 0.133 (0.027) | 0.059 (0.003) | 0.051 (0.007) | 0.153 (0.004) |
| | CMHE ($K = L$) | 0.294 (0.123) | 0.057 (0.005) | 0.061 (0.007) | 0.144 (0.007) |
| | KMeans + TE | **0.895** (0.021) | **0.013** (0.005) | 0.042 (0.010) | **0.020** (0.004) |
| | Virtual Twins | 0.226 (0.069) | 0.045 (0.024) | 0.042 (0.031) | 0.048 (0.011) |

Table 6: Cross-validated performance (with standard deviation in parenthesis) when all covariates clusters presents different treatment responses. This setting aligns with the assumptions made by the step-wise approaches.

### B.4.6 LINEAR TREATMENT RESPONSES

As an alternative setting, we propose a setting with a linear treatment response, as assumed by CMHE Nagpal et al. (2023). To this end, we followed a similar process as described in Section B.1 at the exception of the treatment response. We draw the response under no treatment from a Gompertz, and the one under treatment with the same parameters shifted by the treatment effect. Formally, using the same notation

$$B_z^0 \mid Z = z \sim \text{MVN}(0, I^{10})$$
$$\Gamma_z^0 \mid Z = z \sim \text{MVN}(0, I^{10})$$
$$T_0 \mid Z, X, B_z^0, \Gamma_z^0 = (z, x, \beta_z^0, \gamma_z^0) \sim \text{Gompertz}\left(w_0(\beta_z^0, x), s_0(\gamma_z^0, x)\right)$$

$$T_1 \mid Z, X, B_z^1, \Gamma_z^0, A_z = (z, x, \beta_z^0, \gamma_z^0, a_z) \sim \text{Gompertz}\left(w_0(\beta_z^0, x) \times a_z, s_0(\gamma_z^0, x)\right)$$

with $a_z$ the treatment effect for cluster $z$. In this experiments, we choose $a_0, a_1, a_2 = (0.5, 1, 1.5)$.

Table 7 summarises performances in this scenario. While demonstrating a significant improvement of the CMHE alternatives, CSC presents competing performance on subgroup identification and treatment effect estimates.

| | Model | Rand-Index | IAE$_k(t_{max})$ $k=1$ | $k=2$ | $k=3$ |
|---|---|---|---|---|---|
| **Randomised** | **CSC** | *0.389* (0.065) | 0.033 (0.015) | 0.058 (0.035) | **0.014** (0.006) |
| | **CSC Unadjusted** | **0.411** (0.125) | *0.029* (0.017) | 0.062 (0.021) | *0.017* (0.015) |
| | CMHE ($K=3$) | 0.304 (0.033) | 0.040 (0.003) | *0.050* (0.003) | 0.030 (0.003) |
| | CMHE ($L=3$) | 0.211 (0.109) | 0.051 (0.003) | 0.106 (0.006) | 0.028 (0.008) |
| | CMHE ($K=L$) | 0.384 (0.060) | 0.411 (0.797) | - | 0.435 (0.650) |
| | KMeans + TE | 0.101 (0.139) | 0.080 (0.011) | 0.064 (0.028) | 0.038 (0.023) |
| | Virtual Twins | 0.265 (0.085) | **0.025** (0.006) | **0.018** (0.004) | 0.064 (0.013) |
| **Observational** | **CSC** | 0.311 (0.043) | **0.029** (0.003) | 0.074 (0.028) | 0.027 (0.016) |
| | **CSC Unadjusted** | 0.275 (0.168) | *0.032* (0.005) | *0.044* (0.044) | *0.026* (0.014) |
| | CMHE ($K=3$) | *0.360* (0.068) | 0.038 (0.002) | 0.050 (0.002) | 0.035 (0.009) |
| | CMHE ($L=3$) | 0.285 (0.111) | 0.055 (0.006) | 0.094 (0.007) | **0.017** (0.009) |
| | CMHE ($K=L$) | **0.382** (0.058) | 0.052 (0.011) | 0.128 (0.013) | 0.028 (0.020) |
| | KMeans + TE | 0.115 (0.154) | 0.068 (0.020) | 0.076 (0.031) | 0.031 (0.008) |
| | Virtual Twins | 0.186 (0.117) | 0.048 (0.027) | **0.041** (0.010) | 0.039 (0.012) |

Table 7: Cross-validated performance (with standard deviation in parenthesis) when considering a linear treatment response. This setting aligns with the assumption made by CMHE.

## C  REAL-WORLD CASE STUDY: HETEROGENEITY OF ADJUVANT RADIOTHERAPY RESPONSES IN THE SEER DATASET

To study the medical relevance of the proposed method, we explore data from the Surveillance, Epidemiology, and End Results program[8] (SEER) gathering patients diagnosed with breast cancer between 1992 and 2017. Following Lee et al. (2018); Danks & Yau (2022); Jeanselme et al. (2023), we select women who died from the condition or from cardiovascular diseases. From this observational data, we investigate the impact of adjuvant radiotherapy after chemotherapy on survival outcomes. To this end, we subselect patients with recorded treatment and who received chemotherapy. These criteria led to the selection of 239,855 patients with 22 covariates measured at diagnosis such as diagnosis year, grades, ethnicity, laterality, tumour size and type (see Danks & Yau (2022) for further description).

**Selection of $K$.**  From the selected population, our aim is to identify heterogeneous responses to adjuvant treatment. The first challenge is the selection of the number of groups to use ($K$). We advise to follow medical actionability and consider the change in treatment effects and size of the subgroups when increasing this parameter. In the absence of experts' intuition, one may rely on an elbow rule heuristic over $l_F$. Using Figure 6, the negative log-likelihood presents an elbow for $K = 2$.

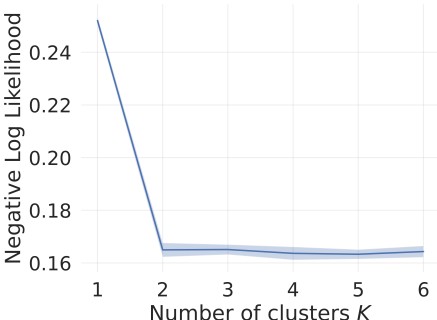

Figure 6: Cross-validated negative log-likelihood as a function of the number of groups ($K$).

**Treatment effect subgroups.**  In this section, we examine the treatment response following adjuvant radiotherapy to identify groups of patients who received surgery and chemotherapy and may benefit from adjuvant radiotherapy. This problem is central to patients' treatment as no evidence-based guidelines for adjuvant therapy exist Lazzari et al. (2023), making this setting more likely to meet the positivity assumption (assumption 6), necessary to study causality in observational data.

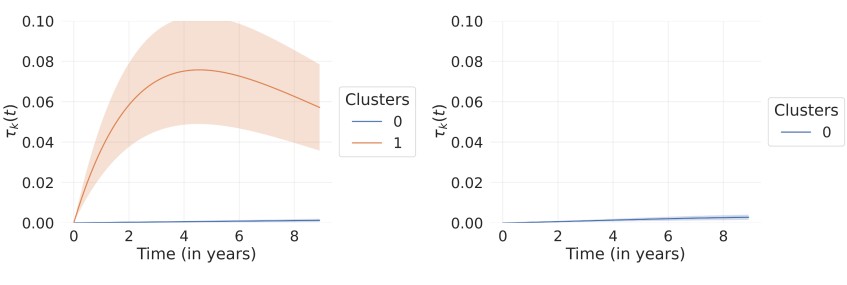

(a) Neural Survival Treatment  (b) Cox Mixture of Heterogeneous Effects

Figure 7: Averaged treatment effect subgroups across 5-fold cross-validation observed in the SEER dataset with the shaded areas representing 95% CI.

---

[8]Available at `https://seer.cancer.gov/`

**Uncovering treatment response.** Figure 7 presents the identified treatment effect subgroups when using CSC and CMHE, with the number of subgroups $K$ selected through hyperparameter tuning. As shown by Figure 6, using the previously described elbow rule leads to the same number of subgroups. As previously mentioned, our proposed methodology presents two strengths that explain the difference in the identified subgroups of treatment effects. First, the survival distribution under treatment is not constrained by the one under the control regime, resulting in more flexible, non-proportional distributions. CMHE's parametrisation, which characterises treatment as a linear shift in the log hazard, results in a proportionality assumption between treated and untreated distributions. Second, CMHE does not account for treatment non-randomisation in its average treatment effect estimation, whereas our use of inverse propensity weighting corrects for any observed ones.

Using a permutation test, we identify as the covariates that most impact the likelihood associated with the model. Figure 8 displays the 10 covariates most indicative of the different treatment response subgroups.

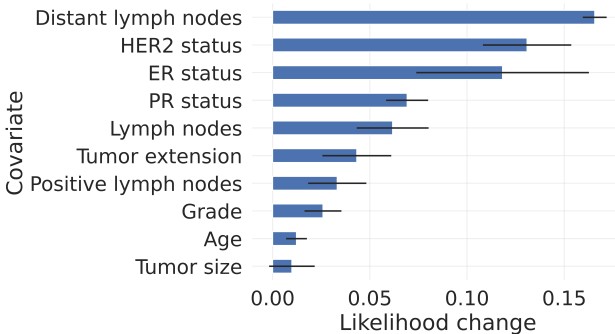

Figure 8: Causal Survival Clustering - Change in log-likelihood given random permutation of a given covariates.

Table 8 summarises the average value across the identified subgroups and the life expectancy gain when using adjuvant radiations measured through the Restricted Mean Survival Time (RMST) Royston & Parmar (2013). Both methodologies identify a population with limited treatment response. However, our proposed methodology identifies a second group, characterised by larger HER2 and larger distant lymph node count, with a positive treatment response, gaining more than half a year of life expectancy over the five years following diagnosis.

| | RMST at 5 years | Population % | Treated % | Distant Lymph Nodes | HER2 Positive | ER Positive |
|---|---|---|---|---|---|---|
| Subgroup 0 | 0.00 (0.00) | 93.3% | 55.5% | 1.18 (5.74) | 17.4% | 46.6% |
| Subgroup 1 | 0.62 (0.16) | 6.7% | 47.5% | 17.75 (16.08) | 23.4% | 48.6% |

Table 8: Causal Survival Clustering subgroups' characteristics in the SEER cohort described through percentage / mean (std).

The proposed analysis pinpoints a group that could benefit from adjuvant radiotherapy. However, our methodologies remain hypothesis-generating tools, requiring further experimental validation, particularly due to potential confounding through hormonal therapy (not available in this dataset), the temporal nature of treatment, and the plurality of treatment options. The quality of available covariates limits this analysis and only serves as an example to medical practitioners to identify subgroups of treatment responses from observational data.

