# OpenReview forum: "Identifying treatment response subgroups in observational time-to-event data"
_ICLR.cc/2025/Conference — ICLR 2025 Conference Withdrawn Submission_

### Official Review · Reviewer_gyV4 · 2024-10-28

**Soundness:** 2
**Presentation:** 2
**Contribution:** 3
**Rating:** 5
**Confidence:** 3

**Summary:**

The study proposes a model to identify subgroups in terms of treatment effects based on observational time-to-event data. The major contribution of this paper is that it is the first work to combine subgroup analysis, treatment effects analysis, and survival analysis together.

**Strengths:**

- The work focuses on model development on the observational data which is vital in this field as clinical trial data are expensive and limited. The study has considered removing potential biases in the data.
- The study models on the survival data, which are prevalent in the medical field, and the proposed method also consider the censoring event.
- The study evaluates the proposed methods in terms of accuracy in subgroup identification and treatment effect estimation, which is solid.
- The study has demonstrated thorough ablation studies, in terms of cluster size, sample size, treatment rate.

**Weaknesses:**

- There was one famous work called "Deepsurv"[1] in this field which was published in 2018. Although it does not consider subgroup identification, but I think you should compare your work with theirs in terms of IAE.
- For identifying the subgroup analysis, the work states that it utilizes a vector $l_k$ (please consider change the notation to $v$ as in your previous sections $l$ is used for log-liklihood), however, there is no description how you got the $l_k$. Is $l_k$ a trainable parameters for your model to learn or some features extracted directly from the baseline table? I would give you a borderline score for now as I didn't get your point at this time but will consider increasing the score if you can explain it clearly in the rebuttal.
- As illustrated in this paper[2], another key information when estimating the treatment effects over time-to-event data is the confidence interval to quantify the how much difference do you observe in the identified subgroup. I think it does matter in the medical science.
- Lastly, I would say the contribution of this work is limited in both methods and application scenario. The methods uses a neural network to model the survival distrbution and estimate the treatment effects is not something new as the work [1] has proposed simiar idea in 2018. The work's framework for addressing the potential bias in observational data is also a classic stragegy used in this field as earliest proposed in "DragonNet"[3]. And in terms of the application scenario, I didn't see any advantage of your method over other external interpretation method as you illustrated in your real-world case study. In your appendix Figure 8, my understanding is that it is very similar to Shapley value like Figure 5 in [2], and it seems that it does not consider the interaction effects based on your interpretation.

[1] Katzman, J. L., Shaham, U., Cloninger, A., Bates, J., Jiang, T., & Kluger, Y. (2018). DeepSurv: personalized treatment recommender system using a Cox proportional hazards deep neural network. BMC medical research methodology, 18, 1-12.

[2] Jiang, L., Xu, C., Bai, Y., Liu, A., Gong, Y., Wang, Y. P., & Deng, H. W. (2024). Autosurv: interpretable deep learning framework for cancer survival analysis incorporating clinical and multi-omics data. NPJ precision oncology, 8(1), 4.

[3] Narendra, K. S., & Mukhopadhyay, S. (1997). Adaptive control using neural networks and approximate models. IEEE Transactions on neural networks, 8(3), 475-485.

**Questions:**

No.

---

### Official Review · Reviewer_9dgA · 2024-10-31

**Soundness:** 2
**Presentation:** 2
**Contribution:** 2
**Rating:** 5
**Confidence:** 3

**Summary:**

The paper introduces a novel neural network-based method for identifying patient subgroups with distinct treatment responses using observational data. This approach tackles the challenges of non-random treatment assignment and censoring, which often introduce biases in observational studies, by incorporating inverse propensity weighting and a monotonic neural network to model complex, non-linear survival distributions under different treatment regimes. Empirical results are presented on both synthetic and real-world datasets.

**Strengths:**

1. Unlike traditional methods that often assume linear treatment responses, the proposed model employs neural networks to allow for non-linear survival functions, making it more adaptable to real-world applications.
2. The integration of monotonic neural networks with IPW is both technically advanced and practically relevant for observational study analysis. This approach helps correct non-random assignment bias without requiring parameterization of the underlying survival distributions.

**Weaknesses:**

1. The main limitation lies in the interpretability of subgroups, especially regarding how these identified subgroups can be understood and applied by clinical practitioners. For example, in Appendix C, it would be helpful to know the characteristics of the two subgroups. If a patient presents with specific features, how would doctors classify them into a subgroup? Additionally, how statistically stable are these subgroups?
2. The presentation of the paper is somewhat tedious. The problem setup section (lines 89 to 289) contains an excess of basic concepts that are likely familiar to scholars in the field, making it less engaging. I recommend condensing the key assumptions and focusing on the unique aspects of the proposed approach.
3. In the synthetic experiment, the number of clusters is known to be three, and the authors use the negative log-likelihood to identify the optimal number of clusters. This heuristic indeed suggests three clusters, aligning with the generative process of the synthetic data. However, in the appendix for the real-world experiment, the cluster number is set to two, which seems questionable, with no verification provided for its accuracy. In summary, the proposed method assumes a pre-determined cluster number, which itself is a complex issue. I am concerned that the performance of the new method may heavily rely on the correctness of this cluster number, and the paper does not provide a robust solution to address this dependency.

**Questions:**

See the weaknesses.

1. Figure 3, line 216: The proposed method uses a separate network W to estimate the propensity score. Could a simple logistic regression suffice instead? I think it’s not always necessary to use complex models—sometimes, simpler approaches work just as well.
2. Figure 7, line 1292: In subfigure (b), there appears to be only one cluster. Is this an error?

---

### Official Review · Reviewer_wkeX · 2024-11-03

**Soundness:** 2
**Presentation:** 2
**Contribution:** 2
**Rating:** 3
**Confidence:** 3

**Summary:**

This paper proposes a new method for identifying subgroups in treatment response estimation for observation time-to-event data. For that, the authors combine ideas from treatment effect estimation adjusting for selection bias, clustering and survival analysis. They further provide experiments on synthetic data and perform an analysis of real-world data.

**Strengths:**

-	The paper addresses an interesting topic in causal inference of clinical relevance going beyond classical CATE estimation to handle subgroup identification and time to event data from RWD / observational data.
-	The combination of the building blocks is comprehensibly explained and mostly well-motivated
-	The authors provide code for reproducibility

**Weaknesses:**

-	The structure of the paper could be clearly improved to clarify the exact research gap, comparison to existent works, and contribution:
    -	In the introduction (line 44-52), method section (l. 201-255), and related work (l.472-479), this paper references mostly CATE methods in the static setting and thus neglecting to explain the connection to CATE methods on survival data (e.g., SurvITE, etc.) or subgroup identification for CATE which makes it hard to judge the novelty of the work.
    -	The potential outcome framework could be introduced more formally and earlier to properly describe the causal nature of the problem
    -	Instead of normal survival forests with clustering, the causal survival forest could be used as a baseline, e.g. also with the R-learner / DoubleML to adjust of selection bias.
    -	The motivation for the subgrouping could be improved, e.g., usually the objective is to get interpretable subgroups
    -	In Eq. (5) and (6), the authors should clearly explain how and why inverse propensity weighting benefits their method, e.g., by properly showing the theoretical benefits under selection bias.
-	The Equations and theoretical motivations seem to be inaccurate and could be improved and extended:
    -	Eq. (4): what exactly is $l_F^*$, what exactly does the “log-likelihood consisting of both factual and counterfactual[...]” represent? Especially wrt to the motivation of IPW later, this is an important part.
    -	What are the consequences and motivation of propensity clipping in Eq.(5) “avoid unstable weights”? (introducing bias but lowering variance, still unbiased but faster convergence?)
    -	Eq. (6) should present an equation or objective and could be better motivated
    -	$u$ should be introduced in Eq. (2) & (6)
- The contribution seems to be mainly incremental, combining existing ideas from subgroup analysis, clustering for survival analysis, and inverse propensity weighting

**Questions:**

-	Since the assumptions seem pretty hard, how robust is the method against possible violations such as of Assumptions 2 and 3?
-	What exactly is the benefit of identifying subgroups compared to normal CATE estimation if they are not interpretable? E.g., in line 1306 a permutation test is used to interpret the subgroups, should such interpretability not be directly incorporated in to the objective?
-	For further questions see weaknesses above.

---

### Official Review · Reviewer_cyGP · 2024-11-03

**Soundness:** 3
**Presentation:** 2
**Contribution:** 2
**Rating:** 5
**Confidence:** 4

**Summary:**

the authors study the problem of identifying subgroups that respond to a treatment differently. they consider a setting with observational study and where the outcomes are right censored. they propose using ML models to flexibly model the nuisance function, and as far as I can tell, this is the main contribution in the paper.

**Strengths:**

Heterogeneity of treatment effects is an important problem.
Authors study this problem in the context of observational studies and under censoring.
The work is well motivated and positioned in the relevant literature.
Although not very novel component-wise, the end-to-end model architecture may be valuable for future work.

**Weaknesses:**

My main criticism toward the work is that it is too assumption heavy. Authors assume observational study is perfect for causal analysis (e.g., no unmeasured confounding) and that the censoring time is conditionally independent of the time-to-event variable. On top of those, the assumption that the number of subgroups is known a priori makes the setting too simple/unrealistic.

While this may not necessarily be a reason for criticism, the results derived in this work follow rather straighforwardly via an amalgamation of the existing work in the literature, bringing the amount of non-incremental contribution made by this work under question, which for me the key takeaway being using nonlinear models for the nuisance functions.

Given this restrictive assumptions, I think it is valuable to evaluate how well the models perform under different violations of the assumptions. The authors could run ablation studies, isolating the violations of different assumptions one at a time to analyze the sensitivity to those assumptions (e.g., no unmeasured confounding and independent censoring and time-to-event).

Authors seem to do this for choosing the number of clusters in Appendix B.3., and to me, there seems to be a non-trivial degrade in performance (larger confidence intervals). This should be more clearly acknowledged in the main text and supported with more discussions in the Appendix for the underlying reasons. In the current form, it seems to be brushed off by saying the model is robust to this, but I do not totally agree.

**Questions:**

Line 155 - I do not think treatment randomization is a "traditional" assumption in the context of observational studies?
Do you perform hyperparameter optimization for the baselines, and if so, where are they described?

---

### Note · Authors · 2024-11-29

**Comment:**

We would like to thank the reviewers for their constructive feedback. We deeply value the suggestions provided and will incorporate them as we revise the paper.

**Withdrawal Confirmation:**

I have read and agree with the venue's withdrawal policy on behalf of myself and my co-authors.